# Extreme bandits

**Alexandra Carpentier**
Statistical Laboratory, CMS
University of Cambridge, UK
a.carpentier@statslab.cam.ac.uk

**Michal Valko**
SequeL team
INRIA Lille - Nord Europe, France
michal.valko@inria.fr

## Abstract

In many areas of medicine, security, and life sciences, we want to allocate limited resources to different sources in order to detect extreme values. In this paper, we study an efficient way to allocate these resources *sequentially* under *limited feedback*. While sequential design of experiments is well studied in *bandit theory*, the most commonly optimized property is the regret with respect to the maximum mean reward. However, in other problems such as network intrusion detection, we are interested in detecting the most extreme value output by the sources. Therefore, in our work we study *extreme regret* which measures the efficiency of an algorithm compared to the oracle policy selecting the source with the *heaviest tail*. We propose the EXTREMEHUNTER algorithm, provide its analysis, and evaluate it empirically on synthetic and real-world experiments.

## 1 Introduction

We consider problems where the goal is to detect *outstanding events* or *extreme values* in domains such as *outlier detection* [1], *security* [18], or *medicine* [17]. The detection of extreme values is important in many life sciences, such as epidemiology, astronomy, or hydrology, where, for example, we may want to know the peak water flow. We are also motivated by *network intrusion detection* where the objective is to find the network node that was compromised, e.g., by seeking the one creating the most number of outgoing connections at once. The search for extreme events is typically studied in the field of *anomaly detection*, where one seeks to find examples that are far away from the majority, according to some problem-specific distance (cf. the surveys [8, 16]).

In anomaly detection research, the concept of anomaly is ambiguous and several definitions exist [16]: point anomalies, structural anomalies, contextual anomalies, etc. These definitions are often followed by heuristic approaches that are seldom analyzed theoretically. Nonetheless, there exist some theoretical characterizations of anomaly detection. For instance, Steinwart et al.[19] consider the level sets of the distribution underlying the data, and rare events corresponding to *rare level sets* are then identified as anomalies. A very challenging characteristic of many problems in anomaly detection is that the data emitted by the sources tend to be *heavy-tailed* (e.g., network traffic [2]) and anomalies come from the sources with the heaviest distribution tails. In this case, rare level sets of [19] correspond to distributions' tails and anomalies to extreme values. Therefore, we focus on the kind of anomalies that are characterized by their *outburst of events* or *extreme values*, as in the setting of [22] and [17].

Since in many cases, the collection of the data samples emitted by the sources is costly, it is important to design *adaptive-learning* strategies that spend more time sampling sources that have a higher risk of being abnormal. The main objective of our work is the *active allocation* of the sampling resources for anomaly detection, in the setting where anomalies are defined as extreme values. Specifically, we consider a variation of the common setting of *minimal feedback* also known as the *bandit setting* [14]: the *learner* searches for the most extreme value that the sources output by probing the sources *sequentially*. In this setting, it must carefully decide which sources to observe

because it only receives the observation from the source it chooses to observe. As a consequence, it needs to allocate the *sampling time* efficiently and should not waste it on sources that do not have an abnormal character. We call this specific setting *extreme bandits*, but it is also known as max-$k$ problem [9, 21, 20]. We emphasize that extreme bandits are poles apart from classical bandits, where the objective is to maximize the sum of observations [3]. An effective algorithm for the classical bandit setting should focus on the source with the highest mean, while an effective algorithm for the extreme bandit problem should focus on the source with the heaviest tail. It is often the case that a heavy-tailed source has a small mean, which implies that the classical bandit algorithms perform poorly for the extreme bandit problem.

The challenging part of our work dwells in the active sampling strategy to detect the heaviest tail under the limited bandit feedback. We proffer EXTREMEHUNTER, a theoretically founded algorithm, that sequentially allocates the resources in an efficient way, for which we prove *performance guarantees*. Our algorithm is efficient under a mild semi-parametric assumption common in *extreme value theory*, while known results by [9, 21, 20] for the extreme bandit problem only hold in a parametric setting (see Section 4 for a detailed comparison).

## 2   Learning model for extreme bandits

In this section, we formalize the *active (bandit) setting* and characterize the measure of performance for any algorithm $\pi$. The *learning setting* is defined as follows. Every time step, each of the $K$ *arms* (sources) emits a sample $X_{k,t} \sim P_k$, unknown to the learner. The precise characteristics of $P_k$ are defined in Section 3. The learner $\pi$ then chooses some arm $I_t$ and then receives only the sample $X_{I_t,t}$. The performance of $\pi$ is evaluated by the most extreme value found and compared to the most extreme value possible. We define the reward of a learner $\pi$ as:

$$G_n^\pi = \max_{t \leq n} X_{I_t,t}$$

The optimal oracle strategy is the one that chooses at each time the arm with the highest potential revealing the highest value, i.e., the arm $*$ with the heaviest tail. Its expected reward is then:

$$\mathbb{E}\left[G_n^*\right] = \max_{k \leq K} \mathbb{E}\left[\max_{t \leq n} X_{k,t}\right]$$

The goal of learner $\pi$ is to get as close as possible to the optimal oracle strategy. In other words, the aim of $\pi$ is to minimize the expected *extreme regret*:

**Definition 1.** *The extreme regret in the bandit setting is defined as:*

$$\mathbb{E}\left[R_n^\pi\right] = \mathbb{E}\left[G_n^*\right] - \mathbb{E}\left[G_n^\pi\right] = \max_{k \leq K} \mathbb{E}\left[\max_{t \leq n} X_{k,t}\right] - \mathbb{E}\left[\max_{t \leq n} X_{I_t,t}\right]$$

## 3   Heavy-tailed distributions

In this section, we formally define our observation model. Let $X_1, \ldots, X_n$ be $n$ i.i.d. observations from a distribution $P$. The behavior of the statistic $\max_{i \leq n} X_i$ is studied by *extreme value theory*. One of the main results is the Fisher-Tippett-Gnedenko theorem [11, 12] that characterizes the limiting distribution of this maximum as $n$ converges to infinity. Specifically, it proves that a rescaled version of this maximum converges to one of the three possible distributions: *Gumbel*, *Fréchet*, or *Weibull*. This rescaling factor depends on $n$. To be concise, we write "$\max_{i \leq n} X_i$ converges to a distribution" to refer to the convergence of the rescaled version to a given distribution. The Gumbel distribution corresponds to the limiting distribution of the maximum of *'not too heavy tailed'* distributions, such as sub-Gaussian or sub-exponential distributions. The Weibull distribution coincides with the behaviour of the maximum of some specific *bounded* random variables. Finally, the Fréchet distribution corresponds to the limiting distribution of the maximum of *heavy-tailed* random variables. As many interesting problems concern heavy-tailed distributions, we focus on Fréchet distributions in this work. The distribution function of a Fréchet random variable is defined for $x \geq m$, and for two parameters $\alpha, s$ as:

$$P(x) = \exp\left\{-\left(\tfrac{x-m}{s}\right)^\alpha\right\}.$$

In this work, we consider positive distributions $P : [0, \infty) \to [0, 1]$. For $\alpha > 0$, the Fisher-Tippett-Gnedenko theorem also states that the statement '$P$ converges to an $\alpha$-Fréchet distribution' is equivalent to the statement '$1 - P$ is a $-\alpha$ regularly varying function in the tail'. These statements are slightly less restrictive than the definition of *approximately* $\alpha$-Pareto distributions[1], i.e., that there exists $C$ such that $P$ verifies:

$$\lim_{x \to \infty} \frac{|1 - P(x) - Cx^{-\alpha}|}{x^{-\alpha}} = 0, \tag{1}$$

or equivalently that $P(x) = 1 - Cx^{-\alpha} + o(x^{-\alpha})$. If and only if $1 - P$ is $-\alpha$ regularly varying in the tail, then the limiting distribution of $\max_i X_i$ is an $\alpha$-Fréchet distribution. The assumption of $-\alpha$ regularly varying in the tail is thus the weakest possible assumption that ensures that the (properly rescaled) maximum of samples emitted by a heavy tailed distributions has a limit. Therefore, the very related assumption of approximate Pareto is almost minimal, but it is (provably) still not restrictive enough to ensure a convergence rate. For this reason, it is natural to introduce an assumption that is slightly stronger than (1). In particular, we assume, as it is common in the extreme value literature, a *second order* Pareto condition also known as the *Hall condition* [13].

**Definition 2.** *A distribution $P$ is $(\alpha, \beta, C, C')$-second order Pareto ($\alpha, \beta, C, C' > 0$) if for $x \geq 0$:*

$$\left| 1 - P(x) - Cx^{-\alpha} \right| \leq C' x^{-\alpha(1+\beta)}$$

By this definition, $P(x) = 1 - Cx^{-\alpha} + \mathcal{O}\left(x^{-\alpha(1+\beta)}\right)$, which is stronger than the assumption $P(x) = 1 - Cx^{-\alpha} + o(x^{-\alpha})$, but similar for small $\beta$.

**Remark 1.** *In the definition above, $\beta$ defines the rate of the convergence (when $x$ diverges to infinity) of the tail of $P$ to the tail of a Pareto distribution $1 - Cx^{-\alpha}$. The parameter $\alpha$ characterizes the heaviness of the tail: The smaller the $\alpha$, the heavier the tail. In the reminder of the paper, we will be therefore concerned with learning the $\alpha$ and identifying the smallest one among the sources.*

## 4 Related work

There is a vast body of research in *offline anomaly detection* which looks for examples that deviate from the rest of the data, or that are not expected from some underlying model. A comprehensive review of many anomaly detection approaches can be found in [16] or [8]. There has been also some work in active learning for anomaly detection [1], which uses a reduction to classification. In *online anomaly detection*, most of the research focuses on studying the setting where a set of variables is monitored. A typical example is the monitoring of cold relief medications, where we are interested in detecting an outbreak [17]. Similarly to our focus, these approaches do not look for outliers in a broad sense but rather for the unusual burst of events [22].

In the extreme values settings above, it is often assumed, that we have *full information* about each variable. This is in contrast to the *limited feedback* or a *bandit setting* that we study in our work. There has been recently some interest in bandit algorithms for heavy-tailed distributions [4]. However the goal of [4] is radically different from ours as they maximize the sum of rewards and not the maximal reward. Bandit algorithms have been already used for network intrusion detection [15], but they typically consider classical or restless setting. [9, 21, 20] were the first to consider the extreme bandits problem, where our setting is defined as the max-$k$ problem. [21] and [9] consider a fully parametric setting. The reward distributions are assumed to be *exactly generalized extreme value distributions*. Specifically, [21] assumes that the distributions are exactly Gumbel, $P(x) = \exp(-(x - m)/s))$, and [9], that the distributions are exactly of Gumbel or Fréchet $P(x) = \exp(-(x - m)^\alpha/(s\alpha)))$. Provided that these assumptions hold, they propose an algorithm for which the regret is asymptotically negligible when compared to the optimal oracle reward. These results are interesting since they are the first for extreme bandits, but their parametric assumption is unlikely to hold in practice and the asymptotic nature of their bounds limits their impact. Interestingly, the objective of [20] is to remove the parametric assumptions of [21, 9] by offering the THRESHOLDASCENT algorithm. However, no analysis of this algorithm for extreme bandits is provided. Nonetheless, to the best of our knowledge, this is the closest competitor for EXTREME-HUNTER and we empirically compare our algorithm to THRESHOLDASCENT in Section 7.

In this paper we also target the extreme bandit setting, but contrary to [9, 21, 20], we only make a semi-parametric assumption on the distribution; the second order Pareto assumption (Definition 2), which is standard in extreme value theory (see e.g., [13, 10]). This is light-years better and significantly weaker than the parametric assumptions made in the prior works for extreme bandits. Furthermore, we provide a *finite-time* regret bound for our more *general semi-parametric setting* (Theorem 2), while the prior works only offer asymptotic results. In particular, we provide an upper bound on the rate at which the regret becomes negligible when compared to the optimal oracle reward (Definition 1).

## 5  Extreme Hunter

In this section, we present our main results. In particular, we present the algorithm and the main theorem that bounds its extreme regret. Before that, we first provide an initial result on the expectation of the maximum of second order Pareto random variables which will set the benchmark for the oracle regret. We first characterize the expectation of the maximum of second order Pareto distributions. The following lemma states that the expectation of the maximum of i.i.d. second order Pareto samples is equal, up to a negligible term, to the expectation of the maximum of i.i.d. Pareto samples. This result is crucial for assessing the benchmark for the regret, in particular the expected value of the maximal oracle sample. Theorem 1 is based on Lemma 3, both provided in the appendix.

**Theorem 1.** *Let $X_1, \ldots, X_n$ be $n$ i.i.d. samples drawn according to $(\alpha, \beta, C, C')$-second order Pareto distribution $P$ (see Definition 2). If $\alpha > 1$, then:*

$$\left| \mathbb{E}(\max_i X_i) - (nC)^{1/\alpha} \Gamma \left(1 - \tfrac{1}{\alpha}\right) \right| \leq \tfrac{4D_2}{n}(nC)^{1/\alpha} + \tfrac{2C'D_{\beta+1}}{C^{\beta+1}n^\beta}(nC)^{1/\alpha} + B = o\left((nC)^{1/\alpha}\right),$$

*where $D_2, D_{1+\beta} > 0$ are some universal constants, and $B$ is defined in the appendix (9).*

Theorem 1 implies that the optimal strategy in hindsight attains the following expected reward:

$$\mathbb{E}\left[G_n^*\right] \approx \max_k \left[(C_k n)^{1/\alpha_k} \Gamma \left(1 - \tfrac{1}{\alpha}\right)\right]$$

Our objective is therefore to find a learner $\pi$ such that $\mathbb{E}\left[G_n^*\right] - \mathbb{E}\left[G_n^\pi\right]$ is negligible when compared to $\mathbb{E}[G_n^*]$, i.e., when compared to $(nC^*)^{1/\alpha^*} \Gamma \left(1 - \tfrac{1}{\alpha^*}\right) \approx n^{1/\alpha^*}$ where $*$ is the optimal arm.

From the discussion above, we know that the minimization of the extreme regret is linked with the identification of the arm with the heaviest tail. Our EXTREMEHUNTER algorithm is based on a classical idea in bandit theory: *optimism in the face of uncertainty*. Our strategy is to estimate $\mathbb{E}\left[\max_{t \leq n} X_{k,t}\right]$ for any $k$ and to pull the arm which maximizes its upper bound. From Definition 2, the estimation of this quantity relies heavily on an efficient estimation of $\alpha_k$ and $C_k$, and on associated confidence widths. This topic is a classic problem in extreme value theory, and such estimators exist provided that one knows a lower bound $b$ on $\beta_k$ [10, 6, 7]. From now on we assume that a constant $b > 0$ such that $b \leq \min_k \beta_k$ is known to the learner. As we argue in Remark 2, this assumption is necessary .

**Algorithm 1** EXTREMEHUNTER

**Input:**
  $K$: number of arms
  $n$: time horizon
  $b$: where $b \leq \beta_k$ for all $k \leq K$
  $N$: minimum number of pulls of each arm
**Initialize:**
  $T_k \leftarrow 0$ for all $k \leq K$
  $\delta \leftarrow \exp(-\log^2 n)/(2nK)$
**Run:**
**for** $t = 1$ **to** $n$ **do**
  **for** $k = 1$ **to** $K$ **do**
    **if** $T_k \leq N$ **then**
      $B_{k,t} \leftarrow \infty$
    **else**
      estimate $\widehat{h}_{k,t}$ that verifies (2)
      estimate $\widehat{C}_{k,t}$ using (3)
      update $B_{k,t}$ using (5) with (2) and (4)
    **end if**
  **end for**
  Play arm $k_t \leftarrow \arg\max_k B_{k,t}$
  $T_{k_t} \leftarrow T_{k_t} + 1$
**end for**

Since our main theoretical result is a *finite-time* upper bound, in the following exposition we carefully describe all the constants and stress what quantities they depend on. Let $T_{k,t}$ be the number of samples drawn from arm $k$ at time $t$. Define $\delta = \exp(-\log^2 n)/(2nK)$ and consider an estimator

$\widehat{h}_{k,t}$ of $1/\alpha_k$ at time $t$ that verifies the following condition with probability $1 - \delta$, for $T_{k,t}$ larger than some constant $N_2$ that depends only on $\alpha_k, C_k, C'$ and $b$:

$$\left| \frac{1}{\alpha_k} - \widehat{h}_{k,t} \right| \leq D\sqrt{\log(1/\delta)}T_{k,t}^{-b/(2b+1)} = B_1(T_{k,t}), \tag{2}$$

where $D$ is a constant that also depends only on $\alpha_k, C_k, C'$, and $b$. For instance, the estimator in [6] (Theorem 3.7) verifies this property and provides $D$ and $N_2$ but other estimators are possible. Consider the associated estimator for $C_k$:

$$\widehat{C}_{k,t} = T_{k,t}^{1/(2b+1)}\left( \frac{1}{T_{k,t}} \sum_{u=1}^{T_{k,t}} \mathbf{1}\left\{ X_{k,u} \geq T_{k,t}^{\widehat{h}_{k,t}/(2b+1)} \right\} \right) \tag{3}$$

For this estimator, we know [7] with probability $1 - \delta$ that for $T_{k,t} \geq N_2$:

$$\left| C_k - \widehat{C}_{k,t} \right| \leq E\sqrt{\log(T_{k,t}/\delta)}\log(T_{k,t})T_{k,T}^{-b/(2b+1)} = B_2(T_{k,t}), \tag{4}$$

where $E$ is derived in [7] in the proof of Theorem 2. Let $N = \max\left( A\log(n)^{2(2b+1)/b}, N_2 \right)$ where $A$ depends on $(\alpha_k, C_k)_k, b, D, E$, and $C'$, and is such that:

$$\max\left( 2B_1(N), 2B_2(N)/C_k \right) \leq 1, \ N \geq (2D\log^2 n)^{(2b+1)/b}, \ \text{and} \ N > \left( \frac{2D\sqrt{\log(n)^2}}{1-\max_k 1/\alpha_k} \right)^{(2b+1)/b}$$

This inspires Algorithm 1, which first pulls each arm $N$ times and then, at each time $t > KN$, pulls the arm that maximizes $B_{k,t}$, which we define as:

$$\left( \left( \widehat{C}_{k,t} + B_2\left( T_{k,t} \right) \right) n \right)^{\widehat{h}_{k,t}+B_1(T_{k,t})} \bar{\Gamma}\left( \widehat{h}_{k,t}, B_1\left( T_{k,t} \right) \right), \tag{5}$$

where $\bar{\Gamma}(x,y) = \tilde{\Gamma}(1 - x - y)$, where we set $\tilde{\Gamma} = \Gamma$ for any $x > 0$ and $+\infty$ otherwise.

**Remark 2.** *A natural question is whether it is possible to learn $\beta_k$ as well. In fact, this is not possible for this model and a negative result was proved by [7]. The result states that in this setting it is not possible to test between two fixed values of $\beta$ uniformly over the set of distributions. Thereupon, we define $b$ as a lower bound for all $\beta_k$. With regards to the Pareto distribution, $\beta = \infty$ corresponds to the exact Pareto distribution, while $\beta = 0$ for such distribution that is not (asymptotically) Pareto.*

We show that this algorithm meets the desired properties. The following theorem states our main result by upper-bounding the extreme regret of EXTREMEHUNTER.

**Theorem 2.** *Assume that the distributions of the arms are respectively $(\alpha_k, \beta_k, C_k, C')$ second order Pareto (see Definition 2) with $\min_k \alpha_k > 1$. If $n \geq Q$, the expected extreme regret of EX-TREMEHUNTER is bounded from above as:*

$$\mathbb{E}\left[ R_n \right] \leq L(nC^*)^{1/\alpha^*}\left( \frac{K}{n}\log(n)^{(2b+1)/b} + n^{-\log(n)(1-1/\alpha^*)} + n^{-b/((b+1)\alpha^*)} \right) = \mathbb{E}\left[ G_n^* \right] o(1),$$

*where $L, Q > 0$ are some constants depending only on $(\alpha_k, C_k)_k, C'$, and $b$ (Section 6).*

Theorem 2 states that the EXTREMEHUNTER strategy performs almost as well as the best (oracle) strategy, up to a term that is negligible when compared to the performance of the oracle strategy. Indeed, the regret is negligible when compared to $(nC^*)^{1/\alpha^*}$, which is the order of magnitude of the performance of the best oracle strategy $\mathbb{E}\left[ G_n^* \right] = \max_{k \leq K} \mathbb{E}\left[ \max_{t \leq n} X_{k,t} \right]$. Our algorithm thus detects the arm that has the heaviest tail.

For $n$ large enough (as a function of $(\alpha_k, \beta_k, C_k)_k, C'$ and $K$), the two first terms in the regret become negligible when compared to the third one, and the regret is then bounded as:

$$\mathbb{E}\left[ R_n \right] \leq \mathbb{E}\left[ G_n^* \right] \mathcal{O}\left( n^{-b/((b+1)\alpha^*)} \right)$$

We make two observations: First, the larger the $b$, the tighter this bound is, since the model is then closer to the parametric case. Second, smaller $\alpha^*$ also tightens the bound, since the best arm is then very heavy tailed and much easier to recognize.

## 6 Analysis

In this section, we prove an upper bound on the extreme regret of Algorithm 1 stated in Theorem 2. Before providing the detailed proof, we give a high-level overview and the intuitions.

In *Step 1*, we define the (favorable) high probability event $\xi$ of interest, useful for analyzing the mechanism of the bandit algorithm. In *Step 2*, given $\xi$, we bound the estimates of $\alpha_k$ and $C_k$, and use them to bound the main upper confidence bound. In *Step 3*, we upper-bound the number of pulls of each suboptimal arm: we prove that with high probability we do not pull them too often. This enables us to guarantee that the number of pulls of the optimal arms $*$ is on $\xi$ equal to $n$ up to a negligible term.

The final *Step 4* of the proof is concerned with using this lower bound on the number of pulls of the optimal arm in order to lower bound the expectation of the maximum of the collected samples. Such step is typically straightforward in the classical (mean-optimizing) bandits by the linearity of the expectation. It is not straightforward in our setting. We therefore prove Lemma 2, in which we show that the expected value of the maximum of the samples in the favorable event $\xi$ will be not too far away from the one that we obtain without conditioning on $\xi$.

**Step 1: High probability event.** In this step, we define the favorable event $\xi$. We set $\delta \overset{\text{def}}{=} \exp(-\log^2 n)/(2nK)$ and consider the event $\xi$ such that for any $k \leq K$, $N \leq T \leq n$:

$$\left| \frac{1}{\alpha_k} - \tilde{h}_k(T) \right| \leq D\sqrt{\log(1/\delta)}T^{-b/(2b+1)},$$

$$\left| C_k - \tilde{C}_k(T) \right| \leq E\sqrt{\log(T/\delta)}T^{-b/(2b+1)},$$

where $\tilde{h}_k(T)$ and $\tilde{C}_k(T)$ are the estimates of $1/\alpha_k$ and $C_k$ respectively using the first $T$ samples. Notice, they are not the same as $\hat{h}_{k,t}$ and $\hat{C}_{k,t}$ which are the estimates of the same quantities at time $t$ for the algorithm, and thus with $T_{k,t}$ samples. The probability of $\xi$ is larger than $1 - 2nK\delta$ by a union bound on (2) and (4).

**Step 2: Bound on $B_{k,t}$.** The following lemma holds on $\xi$ for upper- and lower-bounding $B_{k,t}$.

**Lemma 1.** *(proved in the appendix) On $\xi$, we have that for any $k \leq K$, and for $T_{k,t} \geq N$:*

$$(C_k n)^{\frac{1}{\alpha_k}} \Gamma\left(1 - \frac{1}{\alpha_k}\right) \leq B_{k,t} \leq (C_k n)^{\frac{1}{\alpha_k}} \Gamma\left(1 - \frac{1}{\alpha_k}\right)\left(1 + F\log(n)\sqrt{\log(n/\delta)}T_{k,t}^{-b/(2b+1)}\right) \quad (6)$$

**Step 3: Upper bound on the number of pulls of a suboptimal arm.** We proceed by using the bounds on $B_{k,t}$ from the previous step to upper-bound the number of suboptimal pulls. Let $*$ be the best arm. Assume that at round $t$, some arm $k \neq *$ is pulled. Then by definition of the algorithm $B_{*,t} \leq B_{k,t}$, which implies by Lemma 1:

$$(C^* n)^{1/\alpha^*} \Gamma\left(1 - \frac{1}{\alpha^*}\right) \leq (C_k n)^{1/\alpha_k} \Gamma\left(1 - \frac{1}{\alpha_k}\right)\left(1 + F\log(n)\sqrt{\log(n/\delta)}T_{k,t}^{-b/(2b+1)}\right)$$

Rearranging the terms we get:

$$\frac{(C^* n)^{1/\alpha^*} \Gamma\left(1 - \frac{1}{\alpha^*}\right)}{(C_k n)^{1/\alpha_k} \Gamma\left(1 - \frac{1}{\alpha_k}\right)} \leq 1 + F\log(n)\sqrt{\log(n/\delta)}T_{k,t}^{-b/(2b+1)} \quad (7)$$

We now define $\Delta_k$ which is analogous to the *gap* in the classical bandits:

$$\Delta_k = \frac{(C^* n)^{1/\alpha^*} \Gamma\left(1 - \frac{1}{\alpha^*}\right)}{(C_k n)^{1/\alpha_k} \Gamma\left(1 - \frac{1}{\alpha_k}\right)} - 1$$

Since $T_{k,t} \leq n$, (7) implies for some problem dependent constants $G$ and $G'$ dependent only on $(\alpha_k, C_k)_k, C'$ and $b$, but independent of $\delta$ that:

$$T_{k,t} \leq N + G'\left(\frac{\log^2 n \log(n/\delta)}{\Delta_k^2}\right)^{(2b+1)/(2b)} \leq N + G\left(\log^2 n \log(n/\delta)\right)^{(2b+1)(2b)}$$

This implies that number $T^*$ of pulls of arm $*$ is with probability $1 - \delta'$, at least

$$n - \sum_{k \neq *} G \left( \log^2 n \log(2nK/\delta') \right)^{(2b+1)/(2b)} - KN,$$

where $\delta' = 2nK\delta$. Since $n$ is larger than

$$Q \geq 2KN + 2GK \left( \log^2 n \log \left( 2nK/\delta' \right) \right)^{(2b+1)/(2b)},$$

we have that $T^* \geq \frac{n}{2}$ as a corollary.

**Step 4: Bound on the expectation.** We start by lower-bounding the expected gain:

$$\mathbb{E}[G_n] = \mathbb{E}\left[ \max_{t \leq n} X_{I_t, T_{k,t}} \right] \geq \mathbb{E}\left[ \max_{t \leq n} X_{I_t, T_{k,t}} \mathbf{1}\{\xi\} \right] \geq \mathbb{E}\left[ \max_{t \leq n} X_{*, T_{*,t}} \mathbf{1}\{\xi\} \right] = \mathbb{E}\left[ \max_{i \leq T^*} X_i \mathbf{1}\{\xi\} \right]$$

The next lemma links the expectation of $\max_{t \leq T^*} X_{*,t}$ with the expectation of $\max_{t \leq T^*} X_{*,t} \mathbf{1}\{\xi\}$.

**Lemma 2.** *(proved in the appendix) Let $X_1, \ldots, X_T$ be i.i.d. samples from an $(\alpha, \beta, C, C')$-second order Pareto distribution $F$. Let $\xi'$ be an event of probability larger than $1 - \delta$. Then for $\delta < 1/2$ and for $T \geq Q$ large enough so that $c \max \left( 1/T, 1/T^\beta \right) \leq 1/4$ for a given constant $c > 0$, that depends only on $C, C'$ and $\beta$, and also for $T \geq \log(2) \max \left( C \left( 2C' \right)^{1/\beta}, 8 \log(2) \right)$:*

$$\mathbb{E}\left[ \max_{t \leq T} X_t \mathbf{1}\{\xi\} \right] \geq (TC)^{1/\alpha} \Gamma \left( 1 - \tfrac{1}{\alpha} \right) - \left( 4 + \tfrac{8}{\alpha - 1} \right) (TC)^{1/\alpha} \delta^{1 - 1/\alpha}$$

$$- 2 \left( \tfrac{4D_2}{T} (TC)^{1/\alpha} + \tfrac{2C'D_{1+\beta}}{C^{1+\beta}T^\beta} (TC)^{1/\alpha} + B \right).$$

Since $n$ is large enough so that $2n^2 K\delta' = 2n^2 K \exp \left( -\log^2 n \right) \leq 1/2$, where $\delta' = \exp \left( -\log^2 n \right)$, and the probability of $\xi$ is larger than $1 - \delta'$, we can use Lemma 2 for the optimal arm:

$$\mathbb{E}\left[ \max_{t \leq T^*} X_{*,t} \mathbf{1}\{\xi\} \right] \geq (T^* C^*)^{\frac{1}{\alpha^*}} \left[ \Gamma \left( 1 - \tfrac{1}{\alpha^*} \right) - \left( 4 + \tfrac{8}{\alpha - 1} \right) \delta'^{1 - \frac{1}{\alpha^*}} - \tfrac{8D_2}{T^*} - \tfrac{4C'D_{\max}}{(C^*)^{1+b}(T^*)^b} - \tfrac{2B}{(T^* C^*)^{\frac{1}{\alpha^*}}} \right],$$

where $D_{\max} \overset{\text{def}}{=} \max_i D_{1+\beta_i}$. Using Step 3, we bound the above with a function of $n$. In particular, we lower-bound the last three terms in the brackets using $T^* \geq \frac{n}{2}$ and the $(T^* C^*)^{1/\alpha^*}$ factor as:

$$(T^* C^*)^{1/\alpha^*} \geq (nC^*)^{1/\alpha^*} \left( 1 - \tfrac{GK}{n} \left( \log(2n^2 K/\delta') \right)^{\frac{2b+1}{2b}} - \tfrac{KN}{n} \right)$$

We are now ready to relate the lower bound on the gain of EXTREMEHUNTER with the upper bound of the gain of the optimal policy (Theorem 1), which brings us the upper bound for the regret:

$$\mathbb{E}\left[ R_n \right] = \mathbb{E}\left[ G_n^* \right] - \mathbb{E}\left[ G_n \right] \leq \mathbb{E}\left[ G_n^* \right] - \mathbb{E}\left[ \max_{i \leq T^*} X_i \right] \leq \mathbb{E}\left[ G_n^* \right] - \mathbb{E}\left[ \max_{t \leq T^*} X_{*,t} \mathbf{1}\{\xi\} \right]$$

$$\leq H(nC^*)^{1/\alpha^*} \left( \tfrac{1}{n} + \tfrac{1}{(nC^*)^b} + \tfrac{GK}{n} \left( \log(2n^2 K/\delta') \right)^{\frac{2b+1}{2b}} + \tfrac{KN}{n} + \delta'^{1 - 1/\alpha^*} + \tfrac{B}{(nC^*)^{1/\alpha^*}} \right),$$

where $H$ is a constant that depends on $(\alpha_k, C_k)_k, C'$, and $b$. To bound the last term, we use the definition of $B$ (9) to get the $n^{-\beta^*/((\beta^*+1)\alpha^*)}$ term, upper-bounded by $n^{-b/((b+1)\alpha^*)}$ as $b \leq \beta^*$. Notice that this final term also eats up $n^{-1}$ and $n^{-b}$ terms since $b/((b+1)\alpha^*) \leq \min(1, b)$.

We finish by using $\delta' = \exp \left( -\log^2 n \right)$ and grouping the problem-dependent constants into $L$ to get the final upper bound:

$$\mathbb{E}\left[ R_n \right] \leq L(nC^*)^{1/\alpha^*} \left( \tfrac{K}{n} \log(n)^{(2b+1)/b} + n^{-\log(n)(1 - 1/\alpha^*)} + n^{-b/((b+1)\alpha^*)} \right)$$

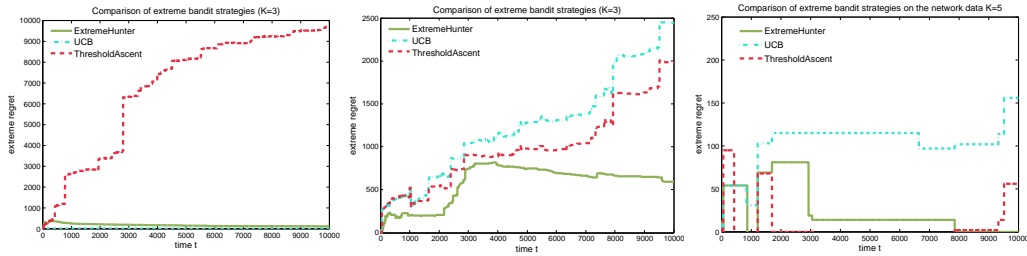

Figure 1: Extreme regret as a function of time for the exact Pareto distributions (left), approximate Pareto (middle) distributions, and the network traffic data (right).

## 7 Experiments

In this section, we empirically evaluate EXTREMEHUNTER on synthetic and real-world data. The measure of our evaluation is the *extreme regret* from Definition 1. Notice that even thought we evaluate the regret as a function of time $T$, the extreme regret is *not cumulative* and it is more in the spirit of *simple regret* [5]. We compare our EXTREMEHUNTER with THRESHOLDASCENT [20]. Moreover, we also compare to classical UCB [3], as an example of the algorithm that aims for the arm with the *highest mean* as opposed to the *heaviest tail*. When the distribution of a single arm has both the highest mean and the heaviest-tail, both EXTREMEHUNTER and UCB are expected to perform the same with respect to the extreme regret. In the light of Remark 2, we set $b = 1$ to consider a wide class of distributions.

**Exact Pareto Distributions**   In the first experiment, we consider $K = 3$ arms with the distributions $P_k(x) = 1 - x^{-\alpha_k}$, where $\alpha = [5, 1.1, 2]$. Therefore, the most heavy-tailed distribution is associated with the arm $k = 2$. Figure 1 (left) displays the averaged result of 1000 simulations with the time horizon $T = 10^4$. We observe that EXTREMEHUNTER eventually keeps allocating most of the pulls to the arm of the interest. Since in this case, the arm with the heaviest tail is also the arm with the largest mean, UCB also performs well and it is even able to detect the best arm earlier. THRESHOLDASCENT, on the other way, was not always able to allocate the pulls properly in $10^4$ steps. This may be due to the discretization of the rewards that this algorithm is using.

**Approximate Pareto Distributions**   For the exact Pareto distributions, the smaller the tail index the higher the mean and even UCB obtains a good performance. However, this is no longer necessarily the case for the approximate Pareto distributions. For this purpose, we perform the second experiment where we mix an exact Pareto distribution with a Dirac distribution in 0. We consider $K = 3$ arms. Two of the arms follow the exact Pareto distributions with $\alpha_1 = 1.5$ and $\alpha_3 = 3$. On the other hand, the second arm has a mixture weight of 0.2 for the exact Pareto distribution with $\alpha_2 = 1.1$ and 0.8 mixture weight of the Dirac distribution in 0. For this setting, the second arm is the most heavy-tailed but the first arms has the largest mean. Figure 1 (middle) shows the result. We see that UCB performs worse since it eventually focuses on the arm with the largest mean. THRESHOLDASCENT performs better than UCB but not as good as EXTREMEHUNTER.

**Computer Network Traffic Data**   In this experiment, we evaluate EXTREMEHUNTER on heavy-tailed network traffic data which was collected from user laptops in the enterprise environment [2]. The objective is to allocate the sampling capacity among the computer nodes (arms), in order to find the largest outbursts of the network activity. This information then serves an IT department to further investigate the source of the extreme network traffic. For each arm, a sample at the time $t$ corresponds to the number of network activity events for 4 consecutive seconds. Specifically, the network events are the starting times of packet flows. In this experiment, we selected $K = 5$ laptops (arms), where the recorded sequences were long enough. Figure 1 (right) shows that EXTREMEHUNTER again outperforms both THRESHOLDASCENT and UCB.

**Acknowledgements**   We would like to thank John Mark Agosta and Jennifer Healey for the network traffic data. The research presented in this paper was supported by Intel Corporation, by French Ministry of Higher Education and Research, and by European Community's Seventh Framework Programme (FP7/2007-2013) under grant agreement n°270327 (CompLACS).

## Footnotes

[1]We recall the definition of the standard Pareto distribution as a distribution $P$, where for some constants $\alpha$ and $C$, we have that for $x \geq C^{1/\alpha}$, $P = 1 - Cx^{-\alpha}$.

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
