[Supplementary Material]

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

# A Proof of Lemma 3

**Lemma 3.** *Assume that $X_1, \ldots, X_T$ are $T$ i.i.d. samples drawn according to $(\alpha, \beta, C, C')$-second order Pareto distribution, then for any $x \geq B$:*

$$\left| \mathbb{P}\left( \max_i X_i \leq x \right) - \exp\left( -TCx^{-\alpha} \right) \right| \leq M \exp\left( -TCx^{-\alpha} \right)$$

$$\textit{where } M = \frac{4}{T}\left( TCx^{-\alpha} \right)^2 + \frac{2C'}{C^{\beta+1}T^\beta}\left( TCx^{-\alpha} \right)^{\beta+1}, \tag{8}$$

*where $B$ is defined as:*

$$B = \max\left( (2C'/C)^{1/(\alpha\beta)}, (8C)^{1/\alpha}, (2TC')^{1/(\alpha(1+\beta))} \right). \tag{9}$$

*Alternatively, let $u \in (0,1)$. When $T \geq \log(1/u)B^\alpha/C$:*

$$\left| \mathbb{P}\left( \max_i X_i \leq (TC/\log(1/u))^{1/\alpha} \right) - u \right| \leq u \left( \frac{4}{T}\log(1/u)^2 + \frac{2C'}{C^{\beta+1}T^\beta}(\log(1/u))^{1+\beta} \right)$$

$$= u \times \mathcal{O}\left( \frac{1}{T}\log(1/u)^2 + \frac{1}{T^\beta}\log(1/u)^{1+\beta} \right)$$

*Proof.* Consider $x \geq B$. Since the samples are i.i.d., we are going to study the following quantity:[2]

$$\mathbb{P}(\max_i X_i \leq x) = P(x)^T \tag{10}$$

Since $P$ is a second order Pareto, we have for any $x \geq 0$:

$$1 - Cx^{-\alpha} - C'x^{-\alpha(1+\beta)} \leq P(x) \leq 1 - Cx^{-\alpha} + C'x^{-\alpha(1+\beta)} \tag{11}$$

Since $x \geq B$, we deduce from the first two terms in (9) that:

$$Cx^{-\alpha} \geq 2C'x^{-\alpha(1+\beta)} \quad \text{and} \quad 2Cx^{-\alpha} \leq 1/4 \tag{12}$$

Let $c_x$ be the quantity that depends on $x$ and that is such that $P(x) = 1 - Cc_x x^{-\alpha}$. With such definition we know by (11) and further by the second inequality in (12) that:

$$|c_x - 1| \leq \frac{C'x^{-\alpha\beta}}{C} \leq 1/2. \tag{13}$$

Let $y = Cc_x x^{-\alpha}$. By (12) and (13) we get that $y \in [0, \frac{1}{2}]$. For any $y \in [0, \frac{1}{2}]$, we have:

$$-y - y^2 \leq \log(1-y) \leq -y$$

Taking the exponential, setting $y = Cc_x x^{-\alpha}$, and raising to the $T$-th power, we obtain:

$$\exp\left( -T\left( Cc_x x^{-\alpha} \right)^2 \right) \leq \frac{(1 - Cc_x x^{-\alpha})^T}{\exp\left( -T\left( Cc_x x^{-\alpha} \right) \right)} \leq 1,$$

which by (10), the definition of $c_x$, and both inequalities in (11) yields:

$$\exp\left( -T\left( 2Cx^{-\alpha} \right)^2 - TC'x^{-\alpha(1+\beta)} \right) \leq \frac{\mathbb{P}(\max_i X_i \leq x)}{\exp(-TCx^{-\alpha})} \leq \exp\left( TC'x^{-\alpha(1+\beta)} \right)$$

After multiplication and subtraction of $\exp(-TCx^{-\alpha})$:

$$\exp\left( -TCx^{-\alpha} \right)\left( \exp\left( -4T(Cx^{-\alpha})^2 - TC'x^{-\alpha(1+\beta)} \right) - 1 \right)$$

$$\leq \mathbb{P}\left( \max_i X_i \leq x \right) - \exp\left( -TCx^{-\alpha} \right)$$

$$\leq \exp\left( -TCx^{-\alpha} \right)\left( \exp\left( TC'x^{-\alpha(1+\beta)} \right) - 1 \right)$$

We will now simplify the $\exp(y) - 1$ terms in the previous inequality. For any $y$ such that $y \in (0, 1/2)$, we have $\exp(y) - 1 \leq 2y$ and for any $y \in \mathbb{R}$ we have that $y \leq \exp(y) - 1$. In particular, this implies whenever $x \geq B \geq (2TC')^{1/(\alpha(1+\beta))}$, which is the third term in (9):

$$\exp\left(-TCx^{-\alpha}\right)\left(-4T(Cx^{-\alpha})^2 - TC'x^{-\alpha(1+\beta)}\right)$$

$$\leq \mathbb{P}\left(\max_i X_i \leq x\right) - \exp\left(-TCx^{-\alpha}\right)$$

$$\leq \exp\left(-TCx^{-\alpha}\right)\left(2TC'x^{-\alpha(1+\beta)}\right)$$

This implies that for any $x \geq B$ and $M$ as defined in (8):

$$\left|P\left(\max_i X_i \leq x\right) - \exp\left(-TCx^{-\alpha}\right)\right| \leq M\exp\left(-TCx^{-\alpha}\right)$$

We now simply reparametrize this upper bound by setting:

$$u = \exp(-TCx^{-\alpha})$$

Then $u \in (0, 1)$ and $x = (TC/\log(1/u))^{1/\alpha}$. Then $x$ is larger than $B$ as soon as $T$ is larger than $\log(1/u)B^{\alpha}/C$. It follows that for such $T$, by the reparametrization in $u$, the rate of convergence of the distribution to a Fréchet distribution:

$$\left|\mathbb{P}\left(\max_i X_i \leq (TC/\log(1/u))^{1/\alpha}\right) - u\right| \leq u\left(\frac{4}{T}(\log 1/u)^2 + \frac{2C'}{C^{\beta+1}T^{\beta}}\log(1/u)^{\beta+1}\right)$$

$\square$

## B   Proof of Theorem 1

**Theorem 1.** *Assume that $X_1, \ldots, X_T$ are $T$ i.i.d. samples drawn according to $(\alpha, \beta, C, C')$-second order Pareto distribution P. If $\alpha > 1$, then:*

$$\left|\mathbb{E}\left(\max_i X_i\right) - (TC)^{1/\alpha}\,\Gamma\left(1-\tfrac{1}{\alpha}\right)\right| \leq \frac{4D_2}{T}(TC)^{1/\alpha} + \frac{2C'D_{\beta+1}}{C^{\beta+1}T^{\beta}}(TC)^{1/\alpha} + B = o\left((TC)^{1/\alpha}\right),$$

*where $D_2 > 0$ and $D_{1+\beta} > 0$ are some universal constants, and $B$ is as defined in (9).*

*Proof.* Since $\alpha > 1$, by definition of a Fréchet distribution:

$$\int_0^\infty \left(1 - \exp(-TCx^{-\alpha})\right)dx = (TC)^{1/\alpha}\,\Gamma\left(1-\tfrac{1}{\alpha}\right) \tag{14}$$

Notice that in (8) we have two terms of the form $\exp(-TCx^{-\alpha})(TCx^{-\alpha})^p$, for $p = 2$ and $p = \beta + 1$. In order to proceed, we first upper-bound the integral of such expression. Through a change of variable (setting $t = TCx^{-\alpha}$) we get that for any $p > 0$:

$$\int_0^\infty \exp(-TCx^{-\alpha})(TCx^{-\alpha})^p dx = \frac{(TC)^{1/\alpha}}{\alpha}\int_0^\infty \exp(-t)t^{p-1-1/\alpha}dt = D_p(TC)^{1/\alpha}, \tag{15}$$

where $D_p = \Gamma(p - 1/\alpha)/\alpha$ is bounded as long as $p > 1/\alpha$, e.g. if $p > 1$. From the definition of expectation we have that:

$$\mathbb{E}\left(\max_i X_i\right) = \int_0^\infty \mathbb{P}\left(\max_i X_i \geq x\right)dx$$

We now bound the difference between this expectation which and the expectation of the Fréchet distribution.

$$\left|\mathbb{E}\left(\max_i X_i\right) - \int_0^\infty \left(1 - \exp\left(-TCx^{-\alpha}\right)\right)dx\right| \leq$$

$$\leq \int_0^\infty 1 - \mathbb{P}\left(\max_i X_i \leq x\right)dx + \int_0^\infty \left(1 - \exp\left(-TCx^{-\alpha}\right)\right)dx$$

$$\leq \left|\int_0^B 1 - \mathbb{P}\left(\max_i X_i \leq x\right)dx + \int_0^B \left(1 - \exp\left(-TCx^{-\alpha}\right)\right)dx\right|$$

$$+ \left|\int_B^\infty 1 - \mathbb{P}\left(\max_i X_i \leq x\right)dx + \int_B^\infty \left(1 - \exp\left(-TCx^{-\alpha}\right)\right)dx\right|,$$

where in the last term we split the domain of integration at $B$. We simply bound the first part by $B$ and for the second term, we use Lemma 3 to obtain:

$$\left| \int_B^\infty 1 - \mathbb{P}\left(\max_i X_i \le x\right) dx + \int_B^\infty \left(1 - \exp\left(-TCx^{-\alpha}\right)\right) dx \right|$$

$$\le \int_0^\infty \left| \mathbb{P}\left(\max_i X_i \le x\right) - \exp\left(-TCx^{-\alpha}\right) \right| dx$$

$$\le \int_0^\infty \exp\left(-TCx^{-\alpha}\right) \left( \frac{4}{T}\left(TCx^{-\alpha}\right)^2 + \frac{2C'}{C^{\beta+1}T^\beta}\left(TCx^{-\alpha}\right)^{\beta+1} \right)$$

Instantiating (15) for $p = 2$ and $p = \beta + 1$, we deduce that:

$$\left| \mathbb{E}(\max_i X_i) - (TC)^{1/\alpha}\Gamma(1-\alpha) \right| \le B + \frac{4D_2}{T}(TC)^{1/\alpha} + \frac{2C'D_{\beta+1}}{C^{\beta+1}T^\beta}(TC)^{1/\alpha}$$

Note that since $\alpha > 1$, we know that $D_{\beta+1}$ and $D_2$ are finite. This concludes the proof. $\quad\square$

## C  Proof of Lemma 1

**Lemma 1.** *On $\xi$, we have that for any $k \le K$, and for $T_{k,t} \ge N$,*

$$(C_k n)^{\frac{1}{\alpha_k}} \Gamma\left(1 - \frac{1}{\alpha_k}\right) \le B_{k,t} \le (C_k n)^{\frac{1}{\alpha_k}} \Gamma\left(1 - \frac{1}{\alpha_k}\right)\left(1 + F\log(n)\sqrt{\log(n/\delta)}T_{k,t}^{-b/(2b+1)}\right) \quad (16)$$

*Proof.* From Step 1, we know that on $\xi$, we can bound $B_{k,t}$ as:

$$(C_k n)^{\frac{1}{\alpha_k}} \Gamma\left(1 - \frac{1}{\alpha_k}\right) \le \left(\left(\widehat{C}_{k,t} + B_2(T_{k,t})\right)n\right)^{\widehat{h}_{k,t} + B_1(T_{k,t})} \bar{\Gamma}\left(\widehat{h}_{k,t}, B_1(T_{k,t})\right)$$

$$\le \left((C_k + 2B_2(T_{k,t}))n\right)^{\frac{1}{\alpha_k} + 2B_1(T_{k,t})} \bar{\Gamma}\left(1/\alpha_k, 2B_1(T_{k,t})\right),$$

since $\Gamma$ is decreasing on $[0, 1]$.

Note that by Theorem 1 we know that $(C_k n)^{1/\alpha_k}\Gamma\left(1 - \frac{1}{\alpha_k}\right)$ is a proxy for the expected maximum of the arm distribution with tail index $\alpha_k$. Factoring our $(C_k n)^{1/\alpha_k}$ we get:

$$\left(\left(C_k + 2B_2\left(T_{k,t}\right)\right)n\right)^{1/\alpha_k + 2B_1(T_{k,t})} \bar{\Gamma}\left(1/\alpha_k, 2B_1(T_{k,t})\right)$$

$$\le (C_k n)^{2B_1(T_{k,t})} \bar{\Gamma}\left(1/\alpha_k, 2B_1\left(T_{k,t}\right)\right) (C_k n)^{\frac{1}{\alpha_k}} \left(1 + \frac{2B_2\left(T_{k,t}\right)}{C_k}\right)^{1/\alpha_k + 2B_1(T_{k,t})} \quad (17)$$

As we pull each arm at least $N$ times (by the assumptions) we have that $T_{k,t} \ge N$ which implies $\max(2B_1(N), 2B_2(N)/C_k) \le 1$. Since $\alpha_k > 1$:

$$\left(1 + \frac{2B_2(T_{k,t})}{C_k}\right)^{1/\alpha_k + 2B_1(T_{k,t})} \le \left(1 + \frac{2E}{C_k}\sqrt{\log(T_{k,t}/\delta)}\log(T_{k,t})T_{k,t}^{-b/(2b+1)}\right)^2$$

$$\le 1 + \frac{6E}{C_k}\sqrt{\log(n/\delta)}\log(n)T_{k,t}^{-b/(2b+1)}. \quad (18)$$

Using again $T_{k,t} \ge N$ and $\log(C_k n)D\sqrt{\log(1/\delta)}N^{-b/(2b+1)} \le 1/2$ for all $k$, we have:

$$(C_k n)^{2B_1(T_{k,t})} = \exp(\log(C_k n)2D\sqrt{\log(1/\delta)}T^{-b/(2b+1)})$$

$$\le 1 + 2\log(C_k n)D\sqrt{\log(1/\delta)}T^{-b/(2b+1)}. \quad (19)$$

Now, let $c$ be the maximum of the absolute value of the derivative of $\Gamma$ on the segment:

$$\left[1 - \max_k \frac{1}{\alpha_k} - D\sqrt{\log(1/\delta)}N^{-b/(2b+1)}, 1 - \min_k \frac{1}{\alpha_k} + D\sqrt{\log(1/\delta)}N^{-b/(2b+1)}\right]$$

Since by the assumption on $N$:

$$N > \left(\frac{2D\sqrt{\log(1/\delta)}}{1 - \max_k 1/\alpha_k}\right)^{(2b+1)/b},$$

we know that $c$ is smaller than the maximum of the absolute value of the derivative of $\Gamma$ function in $\left[\frac{1}{2}(1 - \max_k 1/\alpha_k), \frac{3}{2}(1 - \min_k 1/\alpha_k)\right]$, since $\Gamma$ is a convex and decreasing function on $[0, 1]$. When $\xi$ happens, this implies:

$$\bar{\Gamma}\left(1/\alpha_k, 2B_1(T_{k,t})\right) \leq \Gamma\left(1 - \tfrac{1}{\alpha_k}\right) + 2cB_1(T_{k,t})$$

$$\leq \Gamma\left(1 - \tfrac{1}{\alpha_k}\right) + 2cD\sqrt{\log(1/\delta)}T_{k,t}^{-b/(2b+1)} \tag{20}$$

Finally, combining (17) and (20), we get:

$$((C_k + 2B_2(T_{k,t}))n)^{1/\alpha_k + 2B_1(T_{k,t})}\,\bar{\Gamma}\left(1/\alpha_k, 2B_1(T_{k,t})\right)$$

$$\leq (C_k n)^{1/\alpha_k}\Gamma\left(1 - \tfrac{1}{\alpha_k}\right)\left(1 + F\log(n)\sqrt{\log(n/\delta)}T_{k,t}^{-b/(2b+1)}\right),$$

where $F$ depends on $(\alpha_k, C_k)_k, C', D$, and $E$.

This implies that for $T_{k,t} \geq N$, we can bound $B_{k,t}$ as:

$$(C_k n)^{1/\alpha_k}\,\Gamma\left(1 - \tfrac{1}{\alpha_k}\right) \leq B_{k,t} \leq (C_k n)^{1/\alpha_k}\Gamma\left(1 - \tfrac{1}{\alpha_k}\right)\left(1 + F\log(n)\sqrt{\log(n/\delta)}T_{k,t}^{-b/(2b+1)}\right)$$

$\square$

## D   Proof of Lemma 2

**Lemma 2.** *Let $X_1, \ldots, X_T$ be i.i.d. samples from an $(\alpha, \beta, C, C')$-second order Pareto distribution $F$. Let $\xi'$ be an event of probability larger than $1 - \delta$. Then for $\delta < 1/2$ and for $T \geq Q$ large enough so that $c\max\left(1/T, 1/T^\beta\right) \leq 1/4$ for a given constant $c > 0$, that depends only on $C, C'$ and $\beta$, and also for $T \geq \log(2)\max\left(C\left(2C'\right)^{1/\beta}, 8\log(2)\right)$:*

$$\mathbb{E}\left[\max_{t \leq T} X_t \mathbf{1}\{\xi\}\right] \geq (TC)^{1/\alpha}\,\Gamma\left(1 - \tfrac{1}{\alpha}\right) - \left(4 + \tfrac{8}{\alpha - 1}\right)(TC)^{1/\alpha}\,\delta^{1 - 1/\alpha}$$

$$- 2\left(\tfrac{4D_2}{T}(TC)^{1/\alpha} + \tfrac{2C'D_{1+\beta}}{C^{1+\beta}T^\beta}(TC)^{1/\alpha} + B\right).$$

*Proof.* Since the probability of $\xi'$ is larger than $1 - \delta$:

$$\mathbb{E}\left[\max_{t \leq T} X_t \mathbf{1}\{\xi'\}\right] = \mathbb{E}\left[\max_{t \leq T} X_t\right] - \mathbb{E}\left[\left(\max_{t \leq T} X_t\right)\mathbf{1}\{\xi'^C\}\right]$$

$$= \mathbb{E}\left[\max_{t \leq T} X_t\right] - \int_0^\infty \mathbb{P}\left[\left(\max_{t \leq T} X_t\right)\mathbf{1}\{\xi'^C\} > x\right]dx$$

$$\geq \mathbb{E}\left[\max_{t \leq T} X_t\right] - \int_{x_\delta}^\infty \mathbb{P}\left[\left(\max_{t \leq T} X_t\right) > x\right]dx - \delta x_\delta,$$

where $x_\delta$ is such that $P\left(\max_{t \leq T} X_t \leq x_\delta\right) = 1 - \delta$.

Since we have $T \geq \log(2)\max\left(C\left(2C'\right)^{1/\beta}, 8\log(2)\right)$, and $\delta < 1/2$, we get by Lemma 3:

$$\left|\mathbb{P}\left(\max_i X_i \leq (TC/\log(1/(1-\delta)))^{1/\alpha}\right) - (1-\delta)\right|$$

$$\leq (1-\delta)\left(\tfrac{4}{T}\left(\log\tfrac{1}{1-\delta}\right)^2 + \tfrac{2C'}{C^{1+\beta}}\left(\log\tfrac{1}{1-\delta}\right)^{1+\beta}\right)$$

$$\leq \tfrac{4}{T}(2\delta)^2 + \tfrac{2C'}{C^{1+\beta}}(2\delta)^{1+\beta}$$

$$\leq c\delta\max\left(\tfrac{\delta}{T}, \tfrac{\delta^\beta}{T^\beta}\right)$$

$$\leq c\delta\max\left(\tfrac{1}{T}, \tfrac{1}{T^\beta}\right),$$

where $c > 0$ is a constant that depends only on $C, C'$ and $\beta$. This implies that for $T$ large enough so that $c \max(1/T, 1/T^\beta) \leq 1/4$:

$$\bar{x} = (TC/\log(1/(1-\delta/2)))^{1/\alpha} \geq x_\delta \geq (TC/\log(1/(1-2\delta)))^{1/\alpha} = \underline{x}.$$

By Theorem 1 we can now deduce that:

$$\mathbb{E}\left[\max_{t \leq T} X_t \mathbf{1}\{\xi'\}\right] \geq \mathbb{E}\left[\max_{t \leq T} X_t\right] - \int_{\underline{x}}^\infty \mathbb{P}\left[\left(\max_{t \leq T} X_t\right) > x\right] dx - \delta\bar{x}$$

$$\geq \mathbb{E}\left[\max_{t \leq T} X_t\right] - \int_{\underline{x}}^\infty \left(1 - \exp\left(-TCx^{-\alpha}\right)\right) dx$$

$$- \left(\frac{4D_2}{T}(TC)^{1/\alpha} + \frac{2C'D_{1+\beta}}{C^{1+\beta}T^\beta}(TC)^{1/\alpha} + B\right) - \delta\bar{x}.$$

By the method of substitution, the Taylor expansion and for $\delta$ small enough:

$$\int_{\underline{x}}^\infty \left(1 - \exp\left(-TCx^{-\alpha}\right)\right) dt = \frac{(TC)^{1/\alpha}}{\alpha} \int_0^{\log(1/(1-2\delta))} (1 - \exp(-t)) t^{-1-1/\alpha} dt$$

$$\leq \frac{2(TC)^{1/\alpha}}{\alpha} \int_0^{\log(1/(1-2\delta))} \exp(-t) t^{-1/\alpha} dt$$

$$\leq \frac{2(TC)^{1/\alpha}}{\alpha} \int_0^{\log(1/(1-2\delta))} t^{-1/\alpha} dt$$

$$\leq \frac{2(TC)^{1/\alpha}}{\alpha} \log(1/(1-2\delta))^{1-1/\alpha}$$

$$\leq \frac{8}{\alpha-1}(TC)^{1/\alpha} \delta^{1-1/\alpha}.$$

Next, notice that for small enough $\delta$:

$$\delta\bar{x} \leq 4(TC)^{1/\alpha} \delta^{1-1/\alpha}$$

We get the final lower-bound on $\mathbb{E}\left[\max_{t \leq T} X_t \mathbf{1}\{\xi'\}\right]$ by combining all the above with Theorem 1:

$$(TC)^{1/\alpha} \Gamma\left(1-\tfrac{1}{\alpha}\right) - \left(4+\tfrac{8}{\alpha-1}\right)(TC)^{1/\alpha} \delta^{1-1/\alpha} - 2\left(\frac{4D_2}{T}(TC)^{1/\alpha} + \frac{2C'D_{1+\beta}}{C^{1+\beta}T^\beta}(TC)^{1/\alpha} + B\right).$$

$$\square$$