[Reviews · NeurIPS 2014]

Submitted by Assigned_Reviewer_9

Summary: The paper studied the extreme bandit problem where the goal is to maximize the maximal reward obtained (as opposed to the maximization of the sum of rewards in the classic bandit problem). The authors proposed a new algorithm EXTREMEHUNTER for this task and was able to provide a finite-time performance guarantee. The superiority of their algorithm was demonstrated by experiments on synthetic and real-world data.

Quality: This paper is a solid work. The claims are carefully proved and the technical details are carefully handled (at least in the main text).

Clarity: The paper is clearly written. The problem studied is well motivated by real-life applications and the related work is clearly cited and explained.

Originality: The problem studied is not new. The proposed algorithm is a natural but non-trivial combination of the classic UCB algorithm and the known results from the extreme value theorem. Some of the proof techniques are not trivial though.

Significance: The result is somehow important as it is the first work to provide a finite-time analysis of the extreme bandit problem. However, I am not familiar enough with the extreme bandit literature to be certain.
Summary: The paper is a solid work in that it provides a rigorous finite-time analysis to a complex problem. It should be accepted.

Submitted by Assigned_Reviewer_10

This paper formulates an “extreme bandit” problem. An algorithm sequentially samples from among K distributions with the goal of observing a very extreme (i.e. large) realization. They propose a UCB-type algorithm for this problem, and establish a bound on its “extreme regret.” This bound grows more slowly with the time horizon than the expected value of the most extreme observations of the optimal action, which implies that the algorithm concentrates on the best action asymptotically.

This paper generalizes and strengthens prior results provided by [9,21]. The full parametric assumptions made in those works are replaced by more general assumptions about the tail behavior of reward distributions. In addition, this paper provides finite time rather than asymptotic guarantees. These are valuable extensions, but they feel like incremental progress. I think the paper is slightly below the acceptance threshold.

I don’t find the motivation given in the introduction to be completely satisfying (see below), but in spite of this, it’s possible that there are many very important applications of these techniques with which I am not familiar. (Obviously anomaly detection is very important.) I am somewhat uncertain of my review for this reason.

Mathematical statements in the paper should be more carefully organized and presented in a clearer way. Section 5 tries to simultaneously define the algorithm and introduce a variety of constants that will be used in the theoretical analysis. I recommend defining the algorithm in a clear, self-contained way. You can move some analysis to the appendix if you need more space. Assumptions should be given in the problem formulation, or clearly set aside so the reader can keep track of them. Instead, a key technical assumption is added in the text in lines 213-215. It’s hard to track down all the constant terms, some of which are given only by looking at the paper [6] (right?).

Further comments
- By studying extreme regret, the paper does not precisely focus on identifying the distribution with the heaviest tails so much as on observing large realized values. Obviously these are related objectives, but they are not the same. Why choose to focus on extreme regret?
- What are some natural examples where (1) the goal is to find extreme observations, (2) bandit feedback is a clear obstacle, (3) adaptive sampling is feasible, and (4) observations are costly enough that it’s worth designing very smart sampling schemes.
- In the experiments, which estimator for h_{k,t} did you use?
- The experimental results are averaged over only 100 trials, which seems particularly low given the focus on rare events. You should either increase the number of trials, or include some measure of the variability in the estimates.
- Much of the motivation is that past work makes parametric assumptions that are too restrictive to be practical. It would be nice if the experiments then demonstrated the practical benefit of instead considering a semi-parametric approach.
Summary: This paper generalizes and strengthens past work on "extreme bandit" problems. My impression is that the improvements are incremental instead of groundbreaking, and it's not clear to me that many practical problems are well suited to the proposed algorithm.

Submitted by Assigned_Reviewer_30

The authors fill several missing gaps left by previous papers on a variant of the multi-armed bandit framework, that is the max-k bandit problem, where the goal is to maximize the maximum value returned by the arms. The key contributions of the paper are to generalize (as compared to [9]) the types of distributions that generate the rewards and to introduce an algorithm with theoretical guarantees in the new setting (a significant improvement compared to [20]).

While the algorithmic idea is not surprising (using optimism in the face of uncertainty is a common theme in stochastic optimization), the theoretical analysis is novel, interesting and of possible interest more generally for the bandit literature. In itself, the analysis of the algorithm might be enough for the paper to deserve acceptance.

That being said, I think the write-up should be significantly improved. As a general comment I think the authors focus too much on providing mathematical details in the main text without giving good arguments for their choices. This issue is particularly visible in section 3 and the first half of section 5, where they refer to the extreme value literature to argue for the model choices without giving intuitions about the class of distributions they study.

Regarding the experiments, I have two concerns. The first is that both [9] and [20] did experiments on the same problem (resource-constrained project scheduling) so it would have been very natural to assess the performance of Extreme Hunter on this problem and compare it with the algorithms from these two papers. Why didn’t the authors do this experiment?
The second issue is that I find the results in Figure 1 (left and right) puzzling. For the left figure, the authors mentioned they did 100 repetitions of the experiment. In this context, the sharp discontinuities in the performance of Threshold Ascent are peculiar. The same observation holds for Figure 1 right. Did the authors repeat the 3rd experiment multiple times? If so, how is it the case that the average is not smoother? I am aware the extreme regret is not necessarily monotonous, but I would have expected it to be more smooth with a relatively large number of repetitions of the experiment.

Summary: Given that the theoretical analysis is interesting, I am inclined to recommend acceptance in the hope that the authors can address the issues related to how the results are communicated.
Author Feedback
Author rebuttal: Thank you for the detailed feedback. We appreciate that R2 and R3 deem the work worthy of acceptance and that R2 considers this potentially major.

***R1 (See also R1+R2)

- First, we strongly disagree that our results are only incremental with respect to [9,21] in the 2 aspects mentioned:

A) “simple generalization of the distributions:” Our work applies to most settings where the arms have a limiting distributions for their maximum. On the other hand, [9,21] apply only in a *precise parametric* model (Section 4) that is almost never satisfied in reality (e.g. as illustrated on the network data in Section 7 since - since the real world data are not *exactly* Frechet). Finally and specifically, the algorithms in [9,21] will not work on many common heavy tailed parametric distributions - Cauchy, Student, Gamma, etc

B) “Only finite-time improvement in vs. asymptotic”. We think that finite-time results are important since they analyze behavior in finite running time (we have often only a finite time budget in practice). We believe this finite-time result is a substantial contribution, such as was the finite-time analysis of UCB.

- Why is this important?

The detection of specific anomalies (L46) is very relevant. Though the paper is mostly theoretical, we are directly motivated by detecting anomalies in computer networks (cf. Section 1 and 7). In this case, malicious computers usually produce a low amount of traffic most of the time, but rarely generate very high traffic (e.g. [2]). The problem when malicious computers send extreme values of the traffic is the setting we consider.

Our setting is relevant to many other life problems, where the objective is to detect items that behave most of the time "normally", but that sometimes experience a burst of extreme events [22] - e.g. in situations where the systems are malicious and try to "hide" and in cases of systems that experience moments of extreme instability that are local, but should be detected (e.g. in hydrology).

- Technical

The constants are defined at the same time as the algorithm because they are used in the algorithm.
We will highlight the assumption in L213 - we did not do it because it is natural (and unavoidable - cf. Remark 2). It is in a sense comparable (but not at all the same) to the assumption, in classical bandits, that the arms are bounded/sub-Gaussian.

- Further
-- We focused on the extreme regret because it fits our real world application. The problem you propose is interesting, but is out of the scope of the paper - these two problems are as linked as are the classical bandit and simple regret problem.
-- The computer network scenario (see above).
-- We used the estimator in [6] with second order parameter b.
-- We will increase the number of trials.
-- The experiments show clearly in the approximate Pareto and in the real data that we outperform ThresholdAscent by far.

***R2 (See also R1+R2)

Thank you for your very helpful feedback!

- Experiments

Since we took data from the real network, we believe our experiments are stronger than in [9,20]. We did evaluate [20] in this setting (the results using [9] that we obtained were extremely bad, perhaps due to difficulties in parameter tuning). We can provide more synthetic experiments. For the 1st and 2nd experiment, it is as R2 says. The regret for a single run is non-monotonous and a single instance can cause a jump (for the algorithms that did not "identify" the correct arm). Averaging does "smooth" the curves but not as “fast” as for the classical bandit. Still we can average over much more runs to deliver smoother curves. In practice, few very large scale samples are causing this “behavior” and making “averaging” longer. We only did the 3rd experiment once, since the data were recorded prior and not "sampled" from any model, hence the non-monotonous behaviour of the regret (on a single trajectory). One could think of slicing the data into time slices and averaging over them, but that would be rather artificial. In the full version, we will include more experiments and detail better how we conducted them - in particular, experiments both for expected regret (with many more iterations) and also trajectory-regret on single runs, since they are relevant as well, to illustrate the empirical behavior of the algorithm.

***R1 + R2 about the presentation

We are in a pinch about how to present the analysis: R1 asks us to put more of the analysis in the appendix, R2 asks for more mathematical details in the main text. Since we understand that our analysis is complex, we provided 3 levels of “mathematical involvement”: 1) intuition on L272-285, 2) high-level proof on pages 6-7, and 3) careful treatment in the appendix. We are interested in having a joint opinion of the reviewers.

R1: For clarity, the explicit form of some constant terms are in [6], in order to focus on the core contribution but we can report them precisely in the paper in the final version - we use the constants in the bound for the efficiency of the estimate.

R2: We will improve the writing based on the suggestions and in particular provide more intuition on the class we study (such as in the reply for R1), and make a point on why it is very broad - as it is important point (as you point out).

***R3

We appreciate the encouragement and positive comments! Indeed, while the extreme value theory (EVT) studies the behavior of the heavy tailed distributions, we focus on learning until limited feedback, which is out of scope of EVT. Consequently, the analysis is novel and we hope it inspires more results. Similarly from the bandit side: even though ExtremeHunter is UCB-like, the UCB-like analysis could not be applied and we had to come up with a new style of analysis (cf. e.g., L280-284 and Lemma 2).